# Multi-task learning on partially labeled datasets via invariant/equivariant semi-supervised learning

## Abstract

We investigate the potential of invariant and equivariant semi-supervised learning for addressing the challenges of training multi-task models on partially labeled datasets with differently structured output tasks. Specifically, we use the popular FixMatch method for invariant semi-supervised learning and its equivariant extension Dense FixMatch. We evaluate their performance on the Cityscapes and BDD100K datasets in the context of the prevalent object detection and semantic segmentation tasks in computer vision. We consider varying sizes of the subsets annotated for each task and different overlaps among them. Our results for both invariant and equivariant semi-supervised learning outperform supervised baselines in most situations, with the most significant improvements observed when fewer labeled samples are available for a task and generally better results for the latter approach. Our study suggests that invariant/equivariant learning is a promising general direction for multi-task learning from limited labeled data.

## 1 Introduction

Multi-task learning has emerged as a powerful paradigm in deep learning, with the potential to enable efficient and effective solutions to a range of real-world problems. With the explosion of embedded devices and the growing demand for real-time applications, the ability to solve multiple tasks simultaneously has become a critical requirement. By sharing a common backbone among tasks, multi-task models reduce total system requirements regarding inference-time memory footprint, computation cost, and storage needed for model weights (Vandenhende et al., 2022). Additionally, simultaneously training multiple tasks can improve performance by leveraging task relations or regularizing each other (Caruana, 1997; Bilen & Vedaldi, 2016; Kokkinos, 2017; Zamir et al., 2020). However, multi-task deep learning brings its own challenges, including designing effective architectures and optimizing for multiple objectives, among others (Vandenhende et al., 2022).

This work focuses on the challenges related to the partial annotations available for different tasks in multi-task datasets. Firstly, the number of samples annotated for different tasks can vary greatly, typically depending on the annotation cost of each task. Dense tasks such as semantic segmentation tend to have a per-sample annotation cost orders of magnitude higher than image-level labels (Bearman et al., 2016; Cordts et al., 2016). Other factors, such as the expertise required by the annotators, can also play an important role. For example, annotations from qualified physicians can cost significantly more in medical imaging. In contrast, certain tasks can be annotated at no additional cost, as the loss function can be defined in a self-supervised manner or via other data modalities collected simultaneously. For instance, depth estimation can be performed through stereo image pairs or video sequences (Godard et al., 2017; 2019). As a result, these different costs lead to imbalances between labeled subsets for each task, limited-sized datasets, or high costs to fully annotate all samples.

Furthermore, machine learning projects are iterative, involving multiple cycles of data collection and labeling, training, and testing. At the same time, priorities and requirements often change, and it is common to have different teams working on different tasks. These accentuate the challenge of learning from such data by

making it more likely to have only partially overlapping subsets annotated for each task or possibly fully disjoint datasets.

Learning from such partially labeled datasets using a fully-supervised approach significantly limits the number of samples available for training. One simple solution is to cancel the losses of tasks for samples with no corresponding annotations (Kokkinos, 2017). This approach still under-utilizes data for the less-frequently annotated tasks, leading to suboptimal performance (Nekrasov et al., 2019). Instead, our goal is to involve *all* available samples in learning *all* tasks, hopefully aiding the generalization of all tasks.

Another way to learn from partially labeled multi-task data is by casting it as multiple semi-supervised learning problems solved simultaneously, where from the perspective of each task, there is a set of labeled samples and another set of unlabeled samples. Using semi-supervised learning methods for each task allows the involvement of all data points in the learning of all tasks (Nekrasov et al., 2019; Li & Bilen, 2020; Kim et al., 2018).

We put special emphasis on partially labeled multi-task problems with tasks exhibiting output structures that can be equivariant to certain transformations of the inputs, i.e. equivariant tasks. Typical examples are structured tasks such as the computer vision tasks of object detection and semantic segmentation, produce outputs equivariant to affine transformations of the input such as rotations or translations, but with different structures: segmentation maps vs. bounding boxes.

To this end, we propose *invariant/equivariant learning* as a task-generic semi-supervised learning paradigm that can be adopted for multiple tasks with different output structures simultaneously. Invariant or equivariant learning enforces invariance or equivariance to certain input transformations respectively, a known property of the task, and uses it as a learning objective that can be defined on unlabeled data. We use FixMatch (Sohn et al., 2020a) as the invariant learning approach, using only photometric transformations to allow its use with equivariant tasks, which we refer to as FixMatch*. Dense FixMatch (Martí et al., 2023), which extends FixMatch by using equivariance to affine transformations as a learning objective on unlabeled data, as well as invariance to photometric transformations, is referred throughout the paper as equivariant learning with a slight abuse of language since it does enforce both invariance and equivariance to photometric and affine transformations respectively. Both methods can be applied to multi-task learning on partially labeled datasets in a broader setting where tasks do not share the same output structure, such as semantic segmentation and object detection tasks, while using the core components of modern, state-of-the-art semi-supervised methods for different tasks.

We study the use of invariant/equivariant learning to address the challenges presented by multi-task learning on partially labeled datasets with the Cityscapes (Cordts et al., 2016) and BDD100K (Yu et al., 2020) computer vision datasets using partial annotations for semantic segmentation and object detection tasks.

Our main contributions are:

- We conduct comprehensive experiments representing real-world multi-task learning scenarios by varying annotation amounts for each task, annotation overlaps, and dataset sizes.
- We find that both invariant learning with FixMatch* and equivariant learning with Dense FixMatch generally outperform supervised baselines, more notably for tasks with limited annotations, while the latter approach gives better results for most of the cases we studied.

In light of our findings, we view invariant/equivariant learning as a promising general approach that could be further enhanced when paired with task-specific methods, or other forms of self-supervision when possible.

## 2 Related work

Semi-supervised learning aims to employ unlabeled data alongside smaller amounts of labeled data to improve generalization performance (Yang et al., 2021). Self-training (Lee et al., 2013; Xie et al., 2020b) and consistency regularization (Laine & Aila, 2017; Tarvainen & Valpola, 2017; Miyato et al., 2019) have been the leading approaches to semi-supervised deep learning. Self-training involves using the model's own predictions

on unlabeled data to generate additional labeled data, then used to train the model further. Consistency regularization enforces consistency to small perturbations by using unlabeled samples as targets and minimizing the discrepancy between the model's predictions on different perturbations of the same sample. State-of-the-art solutions combine both and use transformations on the input data as the perturbations (Xie et al., 2020a; Berthelot et al., 2019). In particular, FixMatch (Sohn et al., 2020a) enforces consistency between the predictions of a model on two differently augmented versions of the input, one weakly augmented and one strongly augmented, which we see as using invariance to these transformations as a learning objective on the task outputs that can be applied on unlabeled data and thus we refer to it as *invariant learning*. Similar approaches falling also within the realm of invariant learning have been used in self-supervised representation learning (Chen et al., 2020; Grill et al., 2020) on intermediate features instead. The theoretical analysis of self-training with input consistency regularization in Wei et al. (2021) offers insights into the mechanisms through which these methods have been empirically successful.

Invariance is only desired for some transformations on the input data, but not others, depending on the task. Tasks that predict attributes general to the entire data point are usually invariant tasks, such as image-level classification. In contrast, tasks with structured outputs may be equivariant to certain transformations, meaning that the outputs are transformed in the same way as the inputs. In computer vision tasks such as object detection and semantic segmentation, outputs are equivariant to affine transformations of the input like rotations but remain invariant to photometric transformations like contrast or brightness.

There are various ways to achieve invariant or equivariant models. Firstly, annotations for certain tasks already define this property implicitly. However, it is common to enforce these properties explicitly to help the model generalize since learning them from small amounts of labeled data can be challenging. Data augmentation (Shorten & Khoshgoftaar, 2019; Cubuk et al., 2020) transforms a sample into many versions of it. For invariant tasks, the transformation is applied only to the input. For structured/equivariant tasks, the transformation is applied to both input and output. This way, the model can be guided toward having these properties using the labeled samples. Alternatively, group equivariant convolutional networks (Cohen & Welling, 2016) encode these properties directly into the model architecture by replacing convolutions equivariant only to shifts with group convolutions equivariant to transformations of a particular group, e.g. to affine transformations. However, they are limited to transformations that can be defined as pixel permutations and not on their values.

Instead, invariant or equivariant learning uses invariance or equivariance as a goal to learn from unlabeled data. This not only explicitly enforces the property but also allows training the model with large amounts of otherwise unused samples, which can better cover the input domain.

While most semi-supervised learning algorithms have been developed for image classification, some recent methods focus on specific structured tasks like semantic segmentation (French et al., 2020; Ke et al., 2020; Zou et al., 2021; Yang et al., 2022; Hu et al., 2021; Wang et al., 2022a) or object detection (Jeong et al., 2019; Sohn et al., 2020b; Liu et al., 2021; Xu et al., 2021; Zhou et al., 2022). However, these methods are not universally applicable to all tasks. Pure self-training methods, on the other hand, can be generally used for equivariant tasks but either do not enforce invariance or equivariance (Lee et al., 2013), or rely on separate training, pseudo-labeling and retraining stages (Xie et al., 2020b; Zoph et al., 2020) instead of an online, end-to-end approach. FixMatch (Sohn et al., 2020a) was proposed for image classification but can be used directly for tasks equivariant to affine transformations when discarding such transformations in the augmentation pipeline in order not to break the property of the task, thus enforcing only invariance to the rest of transformations. On the contrary, Dense FixMatch (Martí et al., 2023) enforces both invariance and equivariance to photometric and affine transformations respectively for semi-supervised semantic segmentation, while generating pseudo-labels in an online fashion. These semi-supervised methods, despite being originally designed for a single task, can be applied to other tasks simultaneously. Therefore, we adopt them for semi-supervised multi-task learning as simple instances of invariant and equivariant semi-supervised learning methods respectively.

Different approaches have been proposed to tackle the problem of learning deep multi-task models from partially labeled or disjoint task-specific datasets. In UberNet (Kokkinos, 2017), the losses of a task that a sample is not annotated for are zeroed. The task-related weights are only updated after a sufficient number

of samples annotated for it has been seen. Other works rely on distillation from single-task models trained separately (Nekrasov et al., 2019; Li & Bilen, 2020) or in an alternate training setting (Kim et al., 2018). Adversarial learning has also been used for this purpose (Imran et al., 2020; Perez et al., 2019; Wang et al., 2022b), a semi-supervised learning approach that has fallen out of favor due to the superior performance of consistency regularization and self-training methods (Yang et al., 2021).

Finally, enforcing cross-task consistency between task outputs has been proposed as a way to learn when annotations are missing for a task but can only be used for dense tasks with a shared output structure (Li et al., 2022). Instead, invariant/equivariant learning can be used simultaneously for tasks with differently structured output spaces like image classification, semantic segmentation, and object detection. However, we see invariant/equivariant learning and cross-task consistency as orthogonal and complementary approaches.

## 3 Method

Semi-supervised learning aims to improve the performance of deep neural networks on a specific task by leveraging unlabeled data. Specifically, we aim to train a model on unlabeled samples by explicitly enforcing invariance or equivariance to certain transformations while also training on the task labels when available. While FixMatch (Sohn et al., 2020a) enforces invariance to all input transformations, Dense FixMatch (Martí et al., 2023) allows enforcing equivariance to certain transformations alongside invariance to others.

In particular, FixMatch enforces the model $f$ to produce outputs of differently transformed inputs $\alpha_{inv}(x)$ and $\mathcal{A}_{inv}(x)$ by weak transformations $\alpha_{inv}$ and strong transformations $\mathcal{A}_{inv}$ to be the same:

$$f(\alpha_{inv}(x)) = f(\mathcal{A}_{inv}(x)). \tag{1}$$

In addition, Dense FixMatch enforces the model $f$ to produce outputs from $\alpha_{eq}(x)$ and $\mathcal{A}_{eq}(x)$ such that:

$$(\mathcal{A}_{eq} \circ \alpha_{eq}^{-1})(f(\alpha_{eq}(x))) = f(\mathcal{A}_{eq}(x)), \tag{2}$$

where $\alpha_{eq}$ and $\mathcal{A}_{eq}$ are the particular transformations the task modelled by $f$ is equivariant to. Equation 1 applies to other transformations for which the task is invariant.

When the weak transformations $\alpha_{inv}$ or $\alpha_{eq}$ are the identity, Equations 1 and 2 become the definitions of invariance and equivariance, respectively. This weak transformation is added to avoid over-fitting the pseudo-labels (Sohn et al., 2020a). In the following, we drop the subscript and refer to weak and strong augmentations as simply $\alpha$ and $\mathcal{A}$, including augmentations the tasks are invariant and/or equivariant depending on whether we use FixMatch or Dense FixMatch.

For semi-supervised multi-task learning problems, we explore invariant or equivariant learning using a shared augmentation pipeline for both tasks in $\mathcal{T}$. We obtain outputs for all tasks for weakly and strongly augmented inputs with one forward pass on each. For each task $t \in \mathcal{T}$, we compute a supervised loss $\mathcal{L}_{s,t}$ over the samples $\mathbf{x}$ with ground-truth label $\mathbf{y}_t$, and a consistency or unsupervised loss $\mathcal{L}_{u,t}$ for either the samples not labeled for the task or all samples:

$$\mathcal{L}_{s,t} = \frac{1}{|\mathcal{B}_t^l|} \sum_{(\mathbf{x}_i, \mathbf{y}_i) \in \mathcal{B}_t^l} L_{s,t}(f_t(\alpha(\mathbf{x}_i)), \alpha(\mathbf{y}_i)), \tag{3}$$

$$\mathcal{L}_{u,t} = \frac{1}{|\mathcal{B}_t^u|} \sum_{\mathbf{x}_i \in \mathcal{B}_t^u} L_{u,t}(f_t(\mathcal{A}(\mathbf{x}_i)), (\mathcal{A} \circ \alpha^{-1})(\sigma_t(\bar{f}_t(\alpha(\mathbf{x}_i))))), \tag{4}$$

where $f_t$ is the part of the model that gives outputs for task $t$. $\bar{f}$ is a teacher model with weights following an exponential moving average of the weights of $f$ during training (Tarvainen & Valpola, 2017; Martí et al., 2023). $\sigma_t$ is a task-dependent pseudo-labeling operator that discards low-confidence elements of the model predictions and takes the predictions to the same space as the labels, when applicable. $L_{*,t}$ is the objective for task $t$ defined and averaged over all valid elements of the output structure of the task. $\mathcal{B}_t^l$ is the part of

the mini-batch with annotations for task $t$ with size $|\mathcal{B}_t^l|$, and $\mathcal{B}_t^u$ is the part of the mini-batch without task $t$ labels of size $|\mathcal{B}_t^u|$.

Finally, the global objective for a mini-batch is defined as the weighted sum of the supervised and unsupervised losses for all tasks in $\mathcal{T}$:

$$\mathcal{L} = \sum_{t \in \mathcal{T}} \gamma_t (\mathcal{L}_{s,t} + \lambda_t \mathcal{L}_{u,t}), \tag{5}$$

where $\gamma_t$ and $\lambda_t$ are the task weights controlling the influence of the task over the total loss and the unsupervised loss within the task, respectively.

For our experiments with semantic segmentation and object detection tasks, we set the values for $\gamma_t$ for both tasks to 1. Other approaches for the multi-objective loss in multi-task learning have been explored in the literature and could be used instead (Vandenhende et al., 2022). In addition, we ramp up the importance of the unsupervised loss in the first stages of training when model outputs are unreliable by defining $\lambda_t$ as a sigmoid-shaped function depending on the training step (Tarvainen & Valpola, 2017) but reaching 1 for both tasks.

Both tasks are equivariant to affine transformations and invariant to photometric transformations. Following Sohn et al. (2020a), we use RandAugment (Cubuk et al., 2020) for the strong transformations $\mathcal{A}$ since it applies both types randomly, but only affine transformations are applied to the outputs. For equivariant learning, we use Dense FixMatch with both kinds of transformations. For invariant learning, we discard affine transformations to accommodate our tasks and call this version FixMatch*.

We use the standard cross-entropy objective for semantic segmentation for supervised and unsupervised losses. We do not use any confidence threshold on the pseudo-labels for semantic segmentation, i.e. we use the full segmentation map as a pseudo-target, so the pseudo-labeling operator $\sigma$ is simply the $\arg\max$ operator.

For object detection, we use the focal loss (Lin et al., 2017) for box classification and the generalized IoU loss (Rezatofighi et al., 2019) for box regression. However, picking a point on the precision-recall curve is unavoidable when using predictions as pseudo-labels. It is thus necessary to filter out low-confidence predictions, as is commonly done for inference. Therefore, we discard predicted boxes below a confidence score of 0.5. Earlier in training, we see this as a high threshold leading to high precision but low recall, leading to many false negatives. Accordingly, we modify the unsupervised loss to use only positive targets by discarding all predictions matched to the background.

**Mini-batch sampling.** Since mini-batch sampling significantly impacts the learning dynamics and overall performance in semi-supervised learning, we explore two different approaches following the analysis in Martí et al. (2022). We compare *implicit* sampling, i.e. uniform sampling regardless of the annotations, to *explicit* sampling, i.e. over-sampling of labeled data. In multi-task learning, the latter cannot be applied directly as it requires defining what is a labeled sample. For our experiments, we refer to labeled samples as those that are labeled for at least one task, unless otherwise specified.

## 4 Experiments

To study the effectiveness of invariant or equivariant learning for semi-supervised multi-task learning problem, we focus on training a multi-task model that simultaneously addresses two tasks with distinct output structures: semantic segmentation and object detection, both equivariant tasks to affine transformations of the input. We use FixMatch* as an implementation of invariant learning, using only photometric transformations, and Dense FixMatch for equivariant learning, which also uses affine transformations.

We conduct experiments with two different datasets with available annotations for these two tasks, Cityscapes (Cordts et al., 2016) and BDD100K (Yu et al., 2020), and consider different settings of the problem regarding the available annotations, which cover most cases encountered in real-world datasets. For Cityscapes, we extract the bounding boxes for object detection from dense instance segmentation masks.

First, we study the problem in the low-data regime using the fully labeled part of the Cityscapes dataset. We simulate different amounts of labeled data for each task and different overlaps between the annotated sets. Second, we use the two complete datasets to study real-world cases when more samples are available, each presenting distinct characteristics regarding the available annotations with comparable dataset sizes. We use all annotated samples in Cityscapes and unlabeled samples from the `trainextra` and `sequence` sets. Finally, we experiment on the BDD100K dataset, with subsets annotated for each task that only partially overlap and a significant disparity in the number of labels for each.

**Common settings.** We compare the invariant and equviariant learning approaches to supervised baselines using only the labeled data. In cases where a sample is labeled for only one task, the loss for the other task is zeroed (Kokkinos, 2017). We base our multi-task model on a ResNet-50 (He et al., 2016) backbone and use DeepLabv3+ (Chen et al., 2018) for the semantic segmentation head and RetinaNet (Lin et al., 2017) for the object detection head.

We use a common set of hyper-parameters for all experiments after integrating common values used to train single-task models for these tasks and datasets, except for the number of training steps, which is set differently depending on the dataset. We use random crops at different scales resized to $800 \times 800$ pixels for the supervised baselines and the weak augmentation. For the strong augmentation, we use RandAugment (Cubuk et al., 2020), including both affine and photometric transformations, except for FixMatch* in which case we use only the same photometric transformations. We use a mini-batch size of 8 samples for the supervised baselines. For the explicit sampling setting, we extend the batch with 8 "unlabeled" samples. For the implicit sampling setting, we use the same total batch size as for the explicit setting, 16 samples, where the number of labeled or unlabeled samples is stochastic. We use stochastic gradient descent with momentum, a learning rate of 0.001, and polynomial decay. More details on the hyper-parameters can be found in Appendix A.

**Evaluation.** We perform evaluation with a single forward pass on the full-resolution images of the official `validation` splits. We use as model weights the exponential moving average of the weights obtained during training. We report results using the mean Intersection-over-Union (mIoU) metric for semantic segmentation and mean Average Precision (mAP) for object detection and select the checkpoint with the highest geometric mean of both metrics on the validation set.

**Implementation details.** We use the PyTorch (Paszke et al., 2019) deep learning framework, implement the data augmentation pipelines with Kornia (E. Riba & Bradski, 2020), and run all experiments on a cluster with a variety of Nvidia A40 and A100 GPUs.

## 4.1 Low-data regime experiments on Cityscapes

We simulate multiple annotation scenarios with the fully labeled part of Cityscapes, with 2,975 samples. We randomly drop annotations to keep between 1/32 (93) and 1/2 (1,488) of the samples labeled for each task. We want to simulate multiple situations that could happen in the real world, and annotating samples for the detection task is orders of magnitude cheaper. Thus, it is likely to have significantly more samples annotated for detection. Therefore, when having different sizes for the annotated subsets for each task, the one for detection is larger. We perform experiments for each of the following cases: **(a)** all samples annotated for

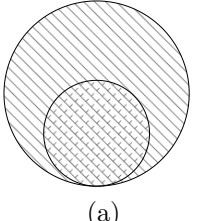 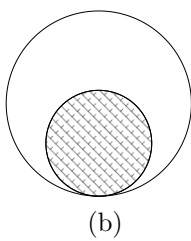 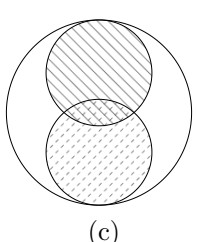 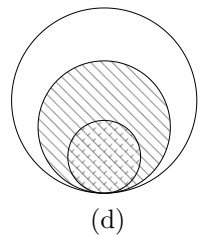 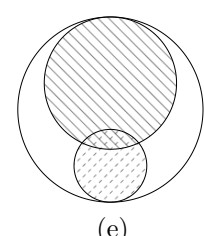

(a)       (b)       (c)       (d)       (e)

Figure 1: Annotated subsets for different tasks in the different low-data regime experiment settings. The hashed area represents the samples labeled for object detection and dashed for semantic segmentation.

Table 1: Results for low-data regime single-task baselines and multi-task experiment (a) on Cityscapes for fully-supervised baselines, and both invariant semi-supervised learning with FixMatch* and equivariant semi-supervised learning with DenseFixMatch. For semi-supervised methods, we compare both the Implicit and Explicit mini-batch sampling approaches. Object detection results are reported as the mAP metric on the `validation` set and semantic segmentation values as the mIoU. For (a) all samples are annotated for detection, # labels refers to the amount annotated for segmentation. In **bold**, the best individual metric for each data setting. □: results for the multi-task model with the highest geometric mean of both task metrics.

| | Method | Sampling | # labels | 93 | 186 | 372 | 744 | 1488 | 2975 |
|---|---|---|---|---|---|---|---|---|---|
| Single-task | Supervised | Uniform | Detection | 0.2010 | 0.2361 | 0.2740 | 0.2982 | 0.2988 | 0.3200 |
| | | | Segmentation | 0.5538 | 0.6243 | 0.6659 | 0.6988 | 0.7288 | 0.7498 |
| | FixMatch* | Implicit | Detection | 0.1781 | 0.2345 | 0.2758 | 0.2957 | 0.3141 | |
| | | | Segmentation | 0.5871 | 0.6543 | 0.7132 | 0.7361 | **0.7523** | |
| | | Explicit | Detection | 0.2358 | 0.2626 | 0.2925 | 0.3093 | 0.3375 | |
| | | | Segmentation | 0.6135 | 0.6698 | 0.7129 | 0.7335 | 0.7494 | |
| | DenseFixMatch | Implicit | Detection | 0.1990 | 0.2405 | 0.2589 | 0.3045 | 0.3300 | |
| | | | Segmentation | 0.6042 | 0.6690 | **0.7175** | **0.7390** | 0.7472 | |
| | | Explicit | Detection | **0.2484** | **0.2732** | **0.2999** | **0.3190** | **0.3382** | |
| | | | Segmentation | **0.6416** | **0.6872** | 0.7145 | 0.7366 | 0.7463 | |
| (a) | Supervised | Uniform | Detection | 0.3173 | 0.3302 | 0.3239 | 0.3205 | 0.3369 | 0.3373 |
| | | | Segmentation | 0.5684 | 0.6449 | 0.6796 | 0.7121 | 0.7369 | 0.7550 |
| | FixMatch* | Implicit | Detection | 0.3407 | 0.3457 | **0.3467** | 0.3504 | 0.3506 | |
| | | | Segmentation | 0.6238 | 0.6811 | 0.7256 | 0.7428 | 0.7498 | |
| | | Explicit | Detection | 0.3253 | 0.3338 | 0.3369 | 0.3448 | 0.3438 | |
| | | | Segmentation | 0.6342 | 0.691 | 0.7218 | 0.7387 | 0.7568 | |
| | DenseFixMatch | Implicit | Detection | **0.3423** | **0.3497** | 0.3440 | **0.3528** | 0.3476 | |
| | | | Segmentation | 0.6160 | 0.6885 | **0.7296** | **0.7464** | **0.7583** | |
| | | Explicit | Detection | 0.3297 | 0.3331 | 0.3383 | 0.3493 | **0.3549** | |
| | | | Segmentation | **0.6453** | **0.7024** | 0.7254 | 0.7421 | 0.7550 | |

detection, different annotated subset sizes for segmentation, **(b)** same annotated subset size for both tasks and complete overlap between them, **(c)** same annotated subset size for both tasks, but random overlap between them, **(d)** different annotated subset size for each task, but subset annotated for segmentation always a subset of that annotated for detection, and **(e)** different annotated subset size for each task, and random overlap between them. Figure 1 illustrates the annotation settings in the experiments. We compare the trained multi-task models against single-task models trained for each task separately, in both the fully-supervised and semi-supervised settings, as well as to the fully-supervised multi-task baselines. Models in experiments (c) and (e) are trained with different data splits due to the random overlap between annotated subsets for each task, so they are only comparable to the fully-labeled single-task and multi-task baselines. We use a training budget of 240 epochs on the `train` set, 89,250 steps.

**Results.** For the single-task baselines in Table 1, either semi-supervised approach improves the results over the supervised baselines for all label and for both tasks settings when using the explicit mini-batch sampling approach. The most significant gains are obtained for the settings with the fewest task labels, and the difference in performance decreases as the number of task labels increases. For implicit mini-batch sampling, semi-supervised learning does not improve the performance of the object detection task, providing benefits only when more labels are available. Generally, Dense FixMatch provides an edge in performance over FixMatch* when fewer task labels are available, hinting at extending with equivariant learning being most important in such cases.

In experiment (a) in Table 1, the explicit mini-batch sampling setting considers samples labeled only for semantic segmentation as labeled and samples them separately from the rest which are only labeled for object detection. This leads to oversampling a portion of the total dataset also for the fully-supervised object detection task. This explains why the implicit setting performs better than explicit sampling across

Table 2: Multi-task experiments (b), with the same annotated subset size for both tasks and full overlap between them, and (c), with random overlap between the subsets of varying sizes.

| | Method | Sampling | # labels | 93 | 186 | 372 | 744 | 1488 |
|---|---|---|---|---|---|---|---|---|
| (b) | Supervised | Uniform | Detection | 0.1988 | 0.2378 | 0.2680 | 0.3020 | 0.3136 |
| | | | Segmentation | 0.5519 | 0.6288 | 0.6736 | 0.7062 | 0.7334 |
| | FixMatch* | Implicit | Detection | 0.1902 | 0.2629 | 0.2866 | 0.3131 | 0.3377 |
| | | | Segmentation | 0.5861 | 0.6789 | 0.7184 | 0.7388 | 0.7491 |
| | | Explicit | Detection | 0.2334 | 0.2787 | **0.3038** | 0.3249 | 0.3376 |
| | | | Segmentation | **0.6325** | 0.6733 | 0.7118 | 0.7395 | 0.7491 |
| | DenseFixMatch | Implicit | Detection | 0.2120 | 0.2517 | 0.2817 | 0.3165 | 0.3370 |
| | | | Segmentation | 0.6002 | 0.6671 | 0.7167 | **0.7437** | **0.7506** |
| | | Explicit | Detection | **0.2570** | **0.2821** | 0.2976 | **0.3279** | **0.3443** |
| | | | Segmentation | 0.6148 | **0.6912** | **0.7233** | 0.7403 | 0.7489 |
| (c) | | | Overlap size | 4 | 18 | 45 | 180 | 753 |
| | Supervised | Uniform | Detection | 0.1895 | 0.2438 | 0.2702 | 0.2958 | 0.3059 |
| | | | Segmentation | 0.5569 | 0.6264 | 0.6702 | 0.7065 | 0.7326 |
| | FixMatch* | Implicit | Detection | 0.1935 | 0.2202 | 0.2906 | 0.3154 | **0.3396** |
| | | | Segmentation | 0.5873 | 0.6712 | 0.7117 | 0.7322 | 0.7504 |
| | | Explicit | Detection | 0.2272 | 0.2661 | 0.2973 | 0.2987 | 0.3391 |
| | | | Segmentation | 0.6044 | 0.6674 | 0.7029 | 0.7228 | **0.7523** |
| | DenseFixMatch | Implicit | Detection | 0.2092 | 0.2579 | 0.2845 | **0.3213** | 0.3364 |
| | | | Segmentation | 0.6047 | **0.6779** | 0.6958 | 0.7278 | 0.7517 |
| | | Explicit | Detection | **0.2493** | **0.2780** | **0.3024** | 0.3177 | 0.3374 |
| | | | Segmentation | **0.6166** | 0.6603 | **0.7065** | **0.7382** | 0.7510 |

Table 3: Multi-task experiments (d), with different annotated subset sizes for each task and full overlap between them, and (e), with different subset sizes and random overlap between them.

| | Method | Sampling | # seg. labels | 93 | | | 186 | | 372 |
|---|---|---|---|---|---|---|---|---|---|
| | | | # det. labels | 372 | 744 | 1488 | 744 | 1488 | 1488 |
| (d) | Supervised | Uniform | Detection | 0.2714 | 0.2951 | 0.3160 | 0.2970 | 0.3141 | 0.3148 |
| | | | Segmentation | 0.5608 | 0.5662 | 0.5797 | 0.6291 | 0.6407 | 0.6788 |
| | FixMatch* | Implicit | Detection | 0.2725 | 0.2995 | 0.3307 | 0.3123 | 0.3257 | 0.3357 |
| | | | Segmentation | 0.5973 | 0.5953 | 0.6042 | 0.6846 | 0.6678 | **0.7252** |
| | | Explicit | Detection | 0.2924 | **0.3185** | 0.3380 | 0.3177 | 0.3370 | 0.3402 |
| | | | Segmentation | 0.6143 | 0.6127 | **0.6119** | 0.6710 | 0.6702 | 0.7177 |
| | DenseFixMatch | Implicit | Detection | 0.2747 | 0.3109 | 0.3378 | 0.3219 | 0.3318 | 0.3339 |
| | | | Segmentation | 0.5967 | 0.6258 | 0.6058 | 0.6784 | **0.6865** | 0.7190 |
| | | Explicit | Detection | **0.2950** | **0.3159** | **0.3408** | **0.3228** | **0.3400** | **0.3413** |
| | | | Segmentation | **0.6191** | **0.6414** | 0.6089 | **0.6882** | 0.6834 | 0.7251 |
| (e) | | | Overlap size | 11 | 20 | 39 | 43 | 85 | 184 |
| | Supervised | Uniform | Detection | 0.2654 | 0.2948 | 0.3221 | 0.2847 | 0.3145 | 0.3121 |
| | | | Segmentation | 0.5826 | 0.6007 | 0.6073 | 0.6411 | 0.6482 | 0.6893 |
| | FixMatch* | Implicit | Detection | 0.2728 | 0.3055 | 0.3081 | 0.2939 | **0.3382** | 0.3356 |
| | | | Segmentation | 0.5948 | 0.6245 | 0.6079 | 0.6713 | **0.6917** | 0.7047 |
| | | Explicit | Detection | 0.2902 | **0.3138** | 0.3343 | 0.3171 | 0.3342 | 0.3455 |
| | | | Segmentation | 0.6106 | **0.6536** | 0.6290 | 0.6593 | 0.6897 | 0.7205 |
| | DenseFixMatch | Implicit | Detection | 0.2853 | 0.3184 | 0.3321 | 0.3107 | 0.3301 | 0.3408 |
| | | | Segmentation | **0.6322** | **0.6360** | 0.6298 | 0.6822 | 0.6737 | **0.7220** |
| | | Explicit | Detection | **0.2960** | **0.3214** | **0.3440** | **0.3231** | **0.3393** | **0.3520** |
| | | | Segmentation | 0.6099 | 0.6207 | **0.6333** | **0.6885** | 0.6714 | 0.7145 |

all but the largest label amounts for object detection. On the contrary, for the semantic segmentation task, explicit sampling gives better results for the settings with the fewest labels. Thus, when considering the joint performance of both tasks as measured by the geometric mean of both task metrics, results are mixed from a sampling approach perspective, but Dense FixMatch gives slightly better performance than FixMatch* in most cases.

In experiments (b) and (c) in Table 2, when having the same amount of labels per task with fixed or random overlap respectively, Dense FixMatch with the explicit setting offers again the best joint performance for most cases, with FixMatch* slightly behind. The performance shows a trend similar to that of the single-task baselines concerning the mini-batch sampling approach used: the explicit setting delivers better results when fewer task labels are available and both offer similar results for the settings with more labels.

In experiment (d) in Table 3, when having different amounts of labels per task but always more detection labels and a fixed overlap, Dense FixMatch with explicit sampling gives the best joint performance for all cases. FixMatch* is only slightly behind and gives the best performance for a specific task in some cases. Again, explicit sampling is most beneficial for the settings with the fewest task labels. In experiment (e), with random overlapping subsets of labeled samples for each task and generally smaller overlapping sizes, the results are similar.

In summary, for the multi-task experiments and across all different data settings, both invariant and equivariant semi-supervised learning with FixMatch* and Dense FixMatch respectively outperform the corresponding supervised baselines in either mini-batch sampling setting when considering joint multi-task performance. Similarly to the results for single-task baselines, the performance gains of semi-supervised learning approaches over supervised learning are most pronounced for the settings with the fewest task labels and decrease with more task labels. The best joint performance is most often obtained using Dense FixMatch and the explicit setting, while FixMatch* generally falls slightly behind.

## 4.2 Experiments on full Cityscapes dataset

In Cityscapes, we have a small `train` set of 2,975 annotated for both tasks, followed by a larger `trainextra` set of 19,998 samples, and a much larger `sequence` set of 86,275 samples but with smaller variability coming from the rest of the frames of the original data collection videos, with no annotations. We use the `validation` set for evaluation. The `trainextra` set has coarse annotations for semantic segmentation obtained in a much cheaper manner by drawing large segmentation blobs but never defining the borders between classes and has been reported to provide little or no value for supervised learning (Luo et al., 2018). Early experiments confirmed this, so we did not use them.

We consider three different cases: using only the `train` set, adding the `trainextra` set, and adding the `sequence` set. We compare computing the unsupervised loss on all samples or the unlabeled ones only. These experiments can be seen as instances of the setting studied in experiment (b) in Section 4.1 where all annotated samples have annotations for both tasks, with labeled to unlabeled data size ratios of approx. 15% and 3% for the `train+trainextra` and `train+trainextra+sequence` experiments, respectively. We use a training budget of 150k steps.

**Results.** Table 4 shows the quantitative results. First, we find that both invariant and equivariant learning with FixMatch* and Dense FixMatch respectively when used on the labeled data only as a regularization term, i.e. without using any extra unlabeled data, improve the results over the supervised baselines. FixMatch* improves object detection mAP by 0.0128 (+3.5%) and semantic segmentation mIoU by 0.0130 (+1.7%). Dense FixMatch gives a slightly larger improvement of 0.0136 (+3.7%) mAP and 0.0160 (+2.1%) mIoU. This shows that enforcing invariance and equivariance properties of the task explicitly is beneficial even when there is no extra unlabeled data available.

Second, adding the unlabeled `trainextra` set and using both FixMatch* and Dense FixMatch as semi-supervised learning methods improves the performance over the supervised multi-task baseline when using the explicit mini-batch sampling approach. When using the implicit mini-batch sampling approach only performance for semantic segmentation is improved, while object detection results are worse than the baseline, which can be due to the low ratio of labeled samples (Martí et al., 2022) and the fact that object detection

Table 4: Experiments on Cityscapes with full labeled set and extra unlabeled data. We compare the *implicit* and *explicit* mini-batch sampling approaches and computing the unsupervised loss $\mathcal{L}_u$ only on unlabeled samples or on all samples. Object detection results are mAP, and semantic segmentation results are mIoU. In **bold**, the best individual metric for each data setting. □: results for the multi-task model with the highest geometric mean of both task metrics.

| Method | Sampling | $\mathcal{L}_u$ applied to | Task | train | +trainextra | +sequence |
|---|---|---|---|---|---|---|
| Supervised | Uniform | | Detection | 0.3666 | | |
| | | | Segmentation | 0.7645 | | |
| FixMatch* | Implicit | Unlabeled only | Detection | | 0.3392 | 0.2769 |
| | | | Segmentation | | 0.7796 | 0.7620 |
| | | All samples | Detection | 0.3794 | 0.3449 | 0.2650 |
| | | | Segmentation | 0.7775 | 0.7807 | 0.7610 |
| | Explicit | Unlabeled only | Detection | | 0.3823 | 0.3839 |
| | | | Segmentation | | **0.7888** | 0.7770 |
| | | All samples | Detection | | **0.3848** | 0.3806 |
| | | | Segmentation | | **0.7857** | 0.7788 |
| DenseFixMatch | Implicit | Unlabeled only | Detection | | 0.3625 | 0.1545 |
| | | | Segmentation | | 0.7820 | 0.7551 |
| | | All samples | Detection | **0.3802** | 0.3505 | 0.2376 |
| | | | Segmentation | **0.7805** | 0.7808 | 0.7593 |
| | Explicit | Unlabeled only | Detection | | 0.3805 | 0.3813 |
| | | | Segmentation | | 0.7876 | **0.7861** |
| | | All samples | Detection | | 0.3798 | **0.3843** |
| | | | Segmentation | | 0.7824 | 0.7806 |

label supervision is more sparse. Computing the unsupervised loss $\mathcal{L}_u$ on unlabeled samples only or on all samples gives similar results. Ultimately, the best joint multi-task performance is obtained for FixMatch* with explicit sampling and $\mathcal{L}_u$ on all samples, improving results over the baseline by 0.0182 (+5.0%) mAP and 0.0212 (+2.8%) mIoU.

Finally, adding the remaining images from the `sequence` set does not further improve the results. For FixMatch*, results are slightly worse than when using only the unlabeled data in the `trainextra` set. For Dense FixMatch, results show no significant improvement. We hypothesize this could be attributed to the low variability of the added images since they correspond to the surrounding frames in the original videos from which the samples in the `train` and `trainextra` sets were extracted. In addition, using implicit mini-batch sampling gives much worse results for both semi-supervised methods, likely due to the even lower ratio of labeled samples.

The qualitative results on the validation set shown in Figure 2 illustrate the differences between the supervised baseline and invariant semi-supervised learning with FixMatch* in either mini-batch sampling setting. Overall, using invariant semi-supervised learning seems to generate more consistent segments for semantic segmentation extending over the full object, with more clear boundaries, as can be seen on poles, traffic signs, and buildings; more clearly for the explicit mini-batch sampling examples. The tree in front of the building in the background of the last example, is wrongly segmented as part of the building behind it, in part or fully. In that same example, another failure case can be seen in which horizontal lines wrongly segmented as poles in the supervised method are more consistently segmented as such for the semi-supervised methods. These can be seen as examples of the known challenge of confirmation bias (Arazo et al., 2020) in semi-supervised learning: some prediction mistakes get exaggerated when reusing model outputs as pseudo-labels. The qualitative differences for object detection are minor. In the first example, the car in the foreground is detected as both a car and a truck for the supervised baseline while the semi-supervised methods detected it only as a car. In the third example, FixMatch* with explicit sampling detects the bikes parked under the trees on the left of the image which are missing for the other two methods, but detects the bike in the foreground as both a bike and a rider, showing some confusion between these classes.

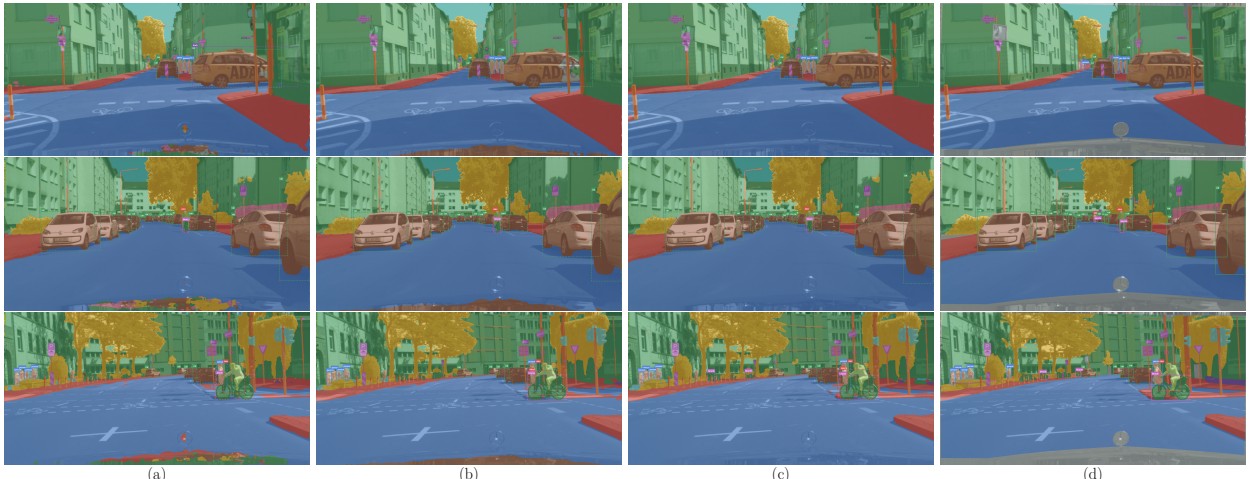

|     (a)      |     (b)      |     (c)      |     (d)      |

Figure 2: Cityscapes qualitative results on the first three samples of the evaluation set. (a) supervised baseline trained on labeled data from `train` set, (b) FixMatch* on `train+trainextra` sets, $\mathcal{L}_u$ on all samples, implicit mini-batch sampling, (c) same as (b) but explicit mini-batch sampling and best overall multi-task model, (d) ground-truth annotations. For object detection predictions, only bounding boxes with confidence over 0.5 are shown. Best seen when zoomed in.

Table 5: Class-wise segmentation IoU on `validation` set of Cityscapes, as well as mean IoU and standard deviation across classes. For supervised baselines and FixMatch* using all samples on `train` and `trainextra` sets for computing $\mathcal{L}_u$ with both *explicit* (E) and *implicit* (I) mini-batch sampling settings. We show in **bold** the best result for each class.

| Method | Road | Side. | Build. | Wall | Fence | Pole | T.light | T.sign | Veg. | Terr. | Sky | Person | Rider | Car | Truck | Bus | Train | Motor. | Bic. | mIoU |
|---|---|---|---|---|---|---|---|---|---|---|---|---|---|---|---|---|---|---|---|---|
| % pixels | 37.65 | 5.41 | 21.92 | 0.73 | 0.82 | 1.48 | 0.20 | 0.67 | 17.32 | 0.83 | 3.35 | 1.30 | 0.22 | 6.51 | 0.30 | 0.39 | 0.11 | 0.08 | 0.71 | |
| Supervised | 0.976 | 0.814 | 0.919 | 0.474 | 0.592 | 0.618 | 0.656 | 0.762 | 0.919 | 0.600 | 0.943 | 0.821 | 0.617 | 0.949 | 0.801 | 0.886 | 0.760 | 0.657 | 0.762 | $0.765_{\pm 0.143}$ |
| FixMatch* (I) | **0.980** | **0.841** | 0.922 | 0.519 | 0.592 | 0.626 | 0.670 | 0.768 | **0.923** | **0.641** | 0.948 | 0.823 | 0.630 | 0.950 | **0.843** | 0.903 | **0.821** | 0.670 | 0.764 | $0.781_{\pm 0.137}$ |
| FixMatch* (E) | **0.980** | 0.840 | **0.924** | **0.571** | **0.603** | **0.635** | **0.679** | **0.779** | 0.922 | 0.604 | **0.949** | **0.830** | **0.639** | **0.953** | 0.842 | **0.907** | 0.810 | **0.687** | **0.773** | $\mathbf{0.786_{\pm 0.132}}$ |

Table 6: Class-wise detection AP on `validation` set of Cityscapes, as well as mean AP and standard deviation across classes. For supervised baselines and FixMatch* using all samples on `train` and `trainextra` sets for computing $\mathcal{L}_u$ with both *explicit* (E) and *implicit* (I) mini-batch sampling settings. We show in **bold** the best result for each class.

| Method | Person | Rider | Car | Truck | Bus | Train | Motor. | Bic. | mAP |
|---|---|---|---|---|---|---|---|---|---|
| % objects | 33.55 | 5.37 | 45.98 | 0.92 | 0.97 | 0.23 | 1.47 | 11.52 | |
| Supervised | **0.360** | 0.400 | 0.525 | 0.325 | 0.558 | 0.241 | 0.234 | 0.289 | $0.367_{\pm 0.114}$ |
| FixMatch* (I) | 0.344 | 0.365 | 0.512 | 0.298 | 0.545 | 0.220 | 0.200 | 0.277 | $0.345_{\pm 0.118}$ |
| FixMatch* (E) | **0.360** | **0.409** | **0.530** | **0.338** | **0.586** | **0.284** | **0.272** | **0.300** | $\mathbf{0.385_{\pm 0.109}}$ |

Table 7: Experiment on BDD100K dataset. We compare the *implicit* and *explicit* mini-batch sampling approaches and computing $\mathcal{L}_u$ on unlabeled or all samples. We provide evaluation metrics and losses for each task. In **bold**, the overall highest or lowest individual metrics or losses respectively. □: results for the multi-task model with the highest geometric mean of both task metrics.

| Method | Sampling | $\mathcal{L}_u$ applied to | Detection | | | Segmentation | |
|---|---|---|---|---|---|---|---|
| | | | mAP | $\mathcal{L}_{cl.}$ | $\mathcal{L}_{reg.}$ | mIoU | $\mathcal{L}_{cl.}$ |
| Supervised, single-task | Uniform, detection only | | **0.2722** | **0.1120** | **0.1708** | - | - |
| | Uniform, segmentation only | | - | - | - | 0.6083 | 0.0253 |
| Supervised, multi-task | Uniform, fully-labeled only | | 0.2200 | 0.2397 | 0.2023 | 0.5770 | 0.0326 |
| | Uniform, partially-labeled | | 0.2686 | 0.1147 | 0.1726 | 0.6047 | 0.0222 |
| FixMatch* | Implicit | Unlabeled only | 0.2564 | 0.1184 | 0.1817 | 0.6318 | 0.0209 |
| | | All samples | 0.2677 | 0.1167 | 0.1777 | 0.6382 | 0.0214 |
| | Explicit | Unlabeled only | 0.2623 | 0.1188 | 0.1800 | 0.6398 | 0.0226 |
| | | All samples | 0.2648 | 0.1189 | 0.1779 | 0.6394 | 0.0228 |
| DenseFixMatch | Implicit | Unlabeled only | 0.2567 | 0.1185 | 0.1818 | 0.6287 | 0.0210 |
| | | All samples | 0.2644 | 0.1172 | 0.1780 | 0.6144 | **0.0206** |
| | Explicit | Unlabeled only | 0.2619 | 0.1183 | 0.1808 | 0.6365 | 0.0224 |
| | | All samples | 0.2670 | 0.1191 | 0.1791 | **0.6470** | 0.0224 |

**Class-wise analysis.** We look at class-wise results for both tasks to better understand the differences between the supervised baseline and the best-performing invariant semi-supervised learning method FixMatch* when using the `trainextra` set, as well as the effect of the two mini-batch sampling approaches considered in the semi-supervised learning setting. Tables 5 and 6 show the class-wise results for the semantic segmentation and the object detection tasks respectively. For both tasks, FixMatch* with the explicit mini-batch sampling approach improves the results for all classes compared to the supervised baseline. Moreover, it reduces the difference between the performance of different classes, as can be seen by the reduced standard deviation of the per-class task metrics over all classes. The previously worse-performing classes get larger improvements. Instead, when using the implicit mini-batch sampling approach, the performance improves for semantic segmentation on all classes but is worse for the object detection task.

## 4.3 Experiments on BDD100K dataset

BDD100K consists of 100,000 videos collected in different locations, weather conditions, and times of the day; and overall offers greater variability than in Cityscapes. It is divided into two datasets according to the tasks that are annotated: (a) the 100K for object detection and other cheap-to-label tasks, such as image-level attributes or lane markings, with one frame annotated per video; and (b) the 10K dataset for semantic segmentation and instance segmentation.

These two datasets only partially overlap, and a small part (454 samples) of the training split of the 10K dataset is used in the validation split of the 100K dataset, which we simply remove from the training split. After merging them, the training set has 69,853 samples annotated for object detection and 6,546 for semantic segmentation, out of which 2,972 samples are annotated for both. There is also a small set of 143 samples not annotated for either. This dataset can be seen as an instance of the setting studied in experiment (e) in Section 4.1. We also merge the validation splits of both datasets for evaluation, forming a validation set with two non-overlapping subsets of 10,000 samples for object detection and 1,000 for semantic segmentation.

We compare the supervised baselines using only the fully-labeled samples (2,972) and the one using all partially-labeled samples (73,427). The mini-batch sampling for the latter is almost identical to the implicit setting since very few samples are fully unlabeled. Here, the explicit setting oversamples the fully-labeled samples. We use an extended training budget of roughly 300k steps.

**Results.** We show the quantitative results in Table 7. Since most of the dataset is labeled for object detection, the supervised baseline that uses all partially-labeled samples is very strong for this task, improving the mAP by 0.0486 (+22.1%) over using only the fully-labeled samples. Still, using a multi-task model causes

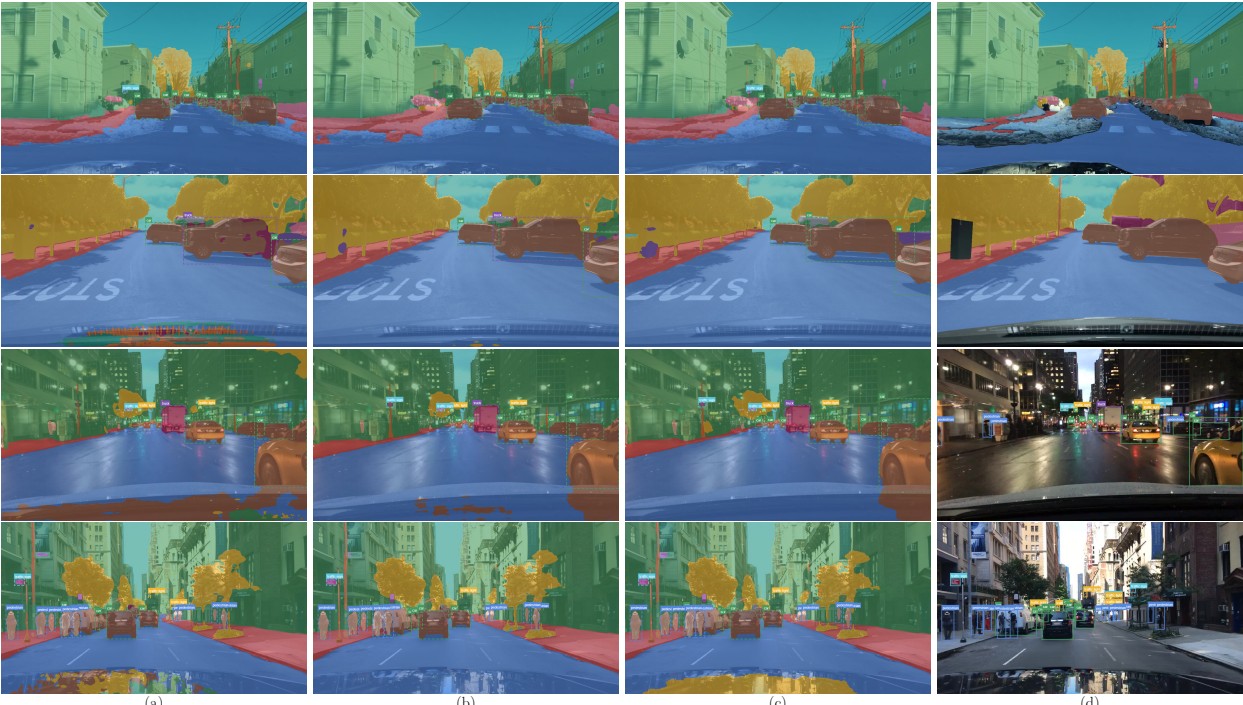

(a)  (b)  (c)  (d)

Figure 3: BDD100K qualitative results on samples from `evaluation` set. (a) supervised baseline trained on labeled data from `train` set, (b) FixMatch*, $\mathcal{L}_u$ on all samples, explicit mini-batch sampling, (c) same as (b) but Dense FixMatch, (d) ground-truth annotations. For object detection predictions, only bounding boxes with confidence over 0.5 are shown. Best seen when zoomed in.

a small drop in performance compared to the single-task baseline of $-0.0036$ mAP (-1.3%). The semantic segmentation task instead is only labeled for a small part of the full dataset so training on all samples labeled for it improves its performance by 0.0277 mIoU (+4.8%), getting it closer to the single-task baseline but still behind by 0.0036 mIoU (-0.6%).

Using semi-supervised invariant or equivariant learning using either mini-batch sampling approach and applying the loss to unlabeled samples or all improves the performance of the semantic segmentation task and hinders object detection. Overall, the best multi-task model is obtained using equivariant learning with Dense FixMatch and explicit mini-batch sampling, applying it to all samples. It achieves a significant improvement over the multi-task baseline trained on partially-labeled samples of 0.0423 mIoU (+7.0%) for the semantic segmentation task but slightly impedes object detection by -0.0016 mAP (-0.6%).

The qualitative results in Figure 3 show the difference between the supervised multi-task baselined trained on all partially labeled samples, and semi-supervised models using the explicit mini-batch sampling approach on all samples for both invariant learning with FixMatch* and equivariant learning with Dense FixMatch. The overall quality of the predictions is clearly worse than for the Cityscapes dataset, likely to the broader domain covered on this dataset, with more varied light and weather conditions, as well as lower quality images. Similarly to Cityscapes, however, the semi-supervised methods tend to generate segmentation masks more consistent within a same object, as can be seen with the fence to the left of the road on the first example, or the better defined boundaries of the poles. On the second example, the predictions for the pick-up truck show some interesting insights. The supervised baseline is not fully segmented as a single object, but with multiple patches of "truck" and "car" classes which are being confused with each other. For the semi-supervised methods, the issue persist but the front part is consistently segmented as "car", the trunk instead is segmented as a "truck". The object detection task gives the class "truck" for two of the methods and "car" for the other, showing no consistency with the segmentation task. The case of pick-up trucks is particular because they don't are not consistently labeled as either class and other similar vehicles like

Table 8: Class-wise segmentation IoU on `validation` set of BDD100K, as well as mean IoU and standard deviation across classes. We show in **bold** the best result for each class and in red the classes that perform significantly worse than the partially-labeled supervised baseline when using semi-supervised methods. Also, mean IoU and standard deviation across classes. The class `train` is not learnt for any of the methods.

| Method | $\mathcal{L}_u$ on | Road | Side. | Build. | Wall | Fence | Pole | T.light | T.sign | Veg. | Terr. | Sky | Person | Rider | Car | Truck | Bus | Train | Motor. | Bic. | mIoU |
|---|---|---|---|---|---|---|---|---|---|---|---|---|---|---|---|---|---|---|---|---|---|
| % pixels | | 24.947 | 2.376 | 17.246 | 0.414 | 0.937 | 1.164 | 0.156 | 0.268 | 17.871 | 1.058 | 20.748 | 0.317 | 0.011 | 10.522 | 1.177 | 0.729 | 0.014 | 0.027 | 0.018 | |
| Fully-labeled | | 0.938 | 0.595 | 0.844 | 0.261 | 0.388 | 0.462 | 0.539 | 0.526 | 0.833 | 0.410 | 0.933 | 0.650 | 0.390 | 0.888 | 0.455 | 0.759 | 0.000 | 0.536 | 0.554 | $0.577_{\pm 0.240}$ |
| Partially-labeled | | 0.946 | 0.632 | 0.860 | 0.355 | 0.552 | 0.485 | 0.575 | 0.552 | 0.861 | 0.465 | 0.953 | 0.636 | 0.327 | 0.909 | 0.621 | 0.795 | 0.000 | 0.477 | 0.487 | $0.605_{\pm 0.238}$ |
| FixMatch*(I) | Unlabeled | **0.953** | **0.673** | 0.870 | 0.418 | 0.553 | 0.506 | 0.589 | 0.583 | 0.872 | 0.517 | 0.958 | 0.662 | 0.461 | 0.919 | 0.661 | **0.859** | 0.000 | 0.438 | 0.513 | $0.632_{\pm 0.232}$ |
| | All | 0.952 | 0.665 | 0.870 | **0.439** | 0.550 | 0.512 | 0.585 | 0.577 | 0.868 | **0.524** | 0.955 | 0.665 | **0.505** | 0.919 | **0.673** | 0.839 | 0.000 | 0.513 | 0.515 | $0.638_{\pm 0.225}$ |
| FixMatch*(E) | Unlabeled | 0.951 | 0.665 | 0.870 | 0.289 | **0.561** | 0.542 | 0.612 | 0.608 | 0.872 | 0.483 | 0.958 | 0.684 | 0.496 | 0.920 | 0.648 | 0.841 | 0.000 | 0.554 | **0.602** | $0.640_{\pm 0.233}$ |
| | All | **0.953** | 0.669 | 0.872 | 0.304 | 0.543 | 0.549 | **0.621** | **0.618** | 0.872 | 0.511 | 0.958 | 0.684 | 0.490 | 0.918 | 0.611 | 0.855 | 0.000 | 0.549 | 0.575 | $0.640_{\pm 0.232}$ |
| DenseFixMatch(I) | Unlabeled | 0.952 | 0.669 | 0.868 | 0.354 | 0.547 | 0.510 | 0.593 | 0.585 | 0.870 | 0.501 | **0.959** | 0.663 | 0.488 | 0.920 | 0.671 | 0.832 | 0.000 | 0.461 | 0.503 | $0.629_{\pm 0.233}$ |
| | All | 0.951 | 0.669 | 0.870 | 0.429 | 0.543 | 0.512 | 0.595 | 0.587 | 0.873 | 0.490 | **0.959** | 0.667 | 0.085 | 0.920 | 0.657 | 0.839 | 0.000 | 0.491 | 0.536 | $0.614_{\pm 0.258}$ |
| DenseFixMatch(E) | Unlabeled | 0.952 | 0.662 | 0.870 | 0.333 | 0.541 | 0.538 | 0.615 | 0.610 | 0.873 | 0.508 | 0.957 | 0.687 | 0.414 | **0.921** | 0.660 | 0.842 | 0.000 | 0.549 | 0.562 | $0.637_{\pm 0.234}$ |
| | All | **0.953** | 0.664 | **0.873** | 0.370 | **0.561** | **0.550** | 0.619 | 0.617 | **0.874** | 0.502 | 0.957 | **0.688** | 0.472 | **0.921** | 0.651 | 0.848 | 0.000 | **0.573** | 0.599 | $\mathbf{0.647_{\pm 0.227}}$ |

Table 9: Class-wise detection AP on `validation` set of BDD100K and mean AP and standard deviation across classes. The partially-labeled supervised baseline performs best for most classes. We show in **bold** the best result for each class and in red the classes that perform significantly worse than the partially-labeled supervised baseline when using semi-supervised methods. The class `train`, the least frequent, is not learnt for any of the methods.

| Method | $\mathcal{L}_u$ on | Pedestrian | Rider | Car | Truck | Bus | Train | Motor. | Bic. | T.light | T.sign | mAP |
|---|---|---|---|---|---|---|---|---|---|---|---|---|
| % objects | | 7.21 | 0.35 | 55.28 | 2.28 | 0.89 | 0.008 | 0.25 | 0.56 | 18.67 | 14.45 | |
| Fully-labeled | | 0.240 | 0.166 | 0.410 | 0.302 | 0.333 | 0.000 | 0.151 | 0.155 | 0.158 | 0.283 | $0.220_{\pm 0.111}$ |
| Partially-labeled | | **0.272** | 0.194 | **0.460** | 0.416 | 0.425 | 0.000 | 0.176 | 0.191 | **0.209** | **0.343** | $\mathbf{0.269_{\pm 0.136}}$ |
| FixMatch*(I) | Unlabeled | 0.259 | 0.185 | 0.438 | 0.405 | 0.412 | 0.000 | 0.172 | 0.174 | 0.199 | 0.321 | $0.257_{\pm 0.131}$ |
| | All | 0.266 | 0.199 | 0.447 | **0.418** | 0.426 | 0.000 | 0.193 | **0.194** | 0.204 | 0.330 | $0.268_{\pm 0.132}$ |
| FixMatch*(E) | Unlabeled | 0.262 | 0.190 | 0.443 | 0.409 | 0.418 | 0.000 | 0.192 | 0.183 | 0.200 | 0.325 | $0.262_{\pm 0.131}$ |
| | All | 0.266 | **0.201** | 0.445 | 0.413 | **0.431** | 0.002 | 0.181 | 0.189 | 0.197 | 0.325 | $0.265_{\pm 0.133}$ |
| DenseFixMatch(I) | Unlabeled | 0.257 | 0.183 | 0.437 | 0.403 | 0.415 | 0.000 | 0.172 | 0.178 | 0.200 | 0.321 | $0.257_{\pm 0.131}$ |
| | All | 0.267 | 0.189 | 0.446 | 0.416 | 0.423 | 0.000 | 0.186 | 0.183 | 0.204 | 0.330 | $0.264_{\pm 0.133}$ |
| DenseFixMatch(E) | Unlabeled | 0.262 | 0.190 | 0.441 | 0.409 | 0.423 | 0.001 | 0.186 | 0.184 | 0.200 | 0.323 | $0.262_{\pm 0.131}$ |
| | All | 0.268 | 0.198 | 0.445 | 0.414 | 0.429 | 0.001 | **0.201** | 0.189 | 0.201 | 0.325 | $0.267_{\pm 0.132}$ |

SUVs are labeled as "car". Moreover, in this particular example the ground truth segmentation wrongly cuts through its trunk labeling it jointly with the background as a "fence". The object detection results are fairly similar, making often the same mistakes and showing only some small differences between methods like the one mentioned for the second example.

**Class-wise analysis.** Tables 8 and 9 show class-wise results for the semantic segmentation and object detection tasks, respectively. We can see the significant class imbalance present for both tasks in the percentage of pixels or objects for each class. The class `Train` is not learnt for either task with any method.

For semantic segmentation, either semi-supervised approach improves the IoU results over the supervised baseline using all partially-labeled samples for most classes. For each method, only up to three classes obtain significantly worse results than for the baseline (values highlighted in red in the table). The two best performing methods deliver results on par or better than the baseline for all classes. Moreover, for all methods except for one, the difference between performance across classes is reduced, meaning that the worse performing classes obtain larger improvements. The exception is Dense FixMatch with the implicit mini-batch sampling setting applied to all samples, which delivers a much worse performance for the rarest class `Rider`. This is likely the effect of severe confirmation bias (Arazo et al., 2020) occurring due to a combination of being the most infrequent class, using the implicit mini-batch sampling approach, and the conceptual similarity of this class with the class `Person`.

On the contrary, for object detection, the semi-supervised methods deliver worse performance for most classes in all possible variations when compared to the supervised baseline trained on all partially-labeled data. In particular they are worse for the most frequent classes in all cases. The few significant gains observed are on the rarest classes, e.g. for `Rider` and `Motorcycle`, leading to a slightly reduced standard deviation of AP across classes for all semi-supervised approaches compared to the baseline.

## 5 Conclusions

We propose invariant/equivariant semi-supervised learning with FixMatch/Dense FixMatch as an effective approach to leverage all samples in multi-task learning from partially labeled datasets, allowing for combinations of tasks with diverse output structures. Extensive experiments across different annotation scenarios, including semantic segmentation and object detection tasks and multiple sampling settings, highlight their superiority over supervised baselines, particularly in tasks with limited labeled samples. Moreover, for our multi-task scenario including two tasks equivariant to affine transformations of the input with different output structures, the combination of invariant and equivariant learning delivers superior performance to invariant learning only in most cases. However, for larger datasets where one task is annotated for most samples, the benefits are observed only for the less frequently annotated task.

Our findings suggest that this approach serves as a promising general direction for this class of problems, which can be enhanced by incorporating task-specific techniques or enforcing cross-task consistency among related tasks when applicable. Future work should also explore and devise mini-batch sampling strategies tailored to the challenges posed by the different scenarios possible in multi-task partially labeled datasets.

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

# A    Hyper-parameter details

Table 10: Hyper-parameter details.

| **Training** | |
|---|---|
| Initial learning rate | 0.001 |
| Decay gamma | 0.9 |
| Nesterov momentum | 0.9 |
| Weight decay | 0.0005 |
| Batch size | Supervised: 8 |
| | Semi-supervised, explicit sampling: 8 + 8 |
| | Semi-supervised, implicit sampling: 16 |
| Model pre-trained on ImageNet | |
| EMA decay | 0.99 |
| Task loss weight $\gamma_t$ | 1 |
| Training steps | 4.1: $\sim$89,250 |
| | 4.2: $\sim$150,000 |
| | 4.3: $\sim$300,000 |
| **Augmentation** | |
| *Common* | |
| Resize scale | 0.5 to 2 |
| Random crop size | 800 x 800 |
| *Normalization* | |
| Channel-wise mean | (0.485, 0.456, 0.406) |
| Channel-wise std. dev. | (0.229, 0.224, 0.225) |
| *Weak* | |
| Horizontal flips* | - |
| *Strong* - RandAugment | |
| # augmentations per batch | 2 |
| AutoContrast | - |
| Equalize | - |
| Invert | - |
| Solarize - threshold | 0.1 to 1 |
| Posterize - bits | 4 to 8 |
| Saturation - range | 0.1 to 1.9 |
| Contrast - range | 0.1 to 1.9 |
| Brightness - range | 0.1 to 1.9 |
| Sharpness - range | 0.1 to 1.9 |
| Shear X/Y* | -17 to 17 |
| Translate X/Y* - % | -15 to 15 |
| Rotate* - degrees | -30 to 30 |
| Horizontal flips* | - |
| Cutout - % of image area | 10 to 30 |
| * disabled for invariant only FixMatch | |
| **FixMatch/DenseFixMatch** | |
| Pseudo-label confidence threshold $\sigma_t$ | Detection: 0.5 |
| | Segmentation: 0 |
| Unsupervised loss weight $\lambda_t$ | 1 |
| Unsupervised loss warmup steps | 4.1: $\sim$5% total training steps |
| | 4.2,4.3: $\sim$10% total training steps |