# OpenReview forum: "Multi-task learning on partially labeled datasets via invariant/equivariant semi-supervised learning"
_TMLR — Rejected by TMLR_

### Review · Reviewer_QMGY · 2024-02-26

**Summary Of Contributions:**

This paper uses invariant/equivariant semi-supervised learning with FixMatch/Dense FixMatch as a task-generic semi-supervised learning paradigm that can be adopted for multiple tasks with different output structures simultaneously.

The authors conduct comprehensive experiments on the Cityscapes and BDD100K for semantic segmentation and object detection tasks. They find that both invariant learning and equivariant learning outperform supervised baselines, more notably for tasks with limited annotations.

**Audience:**

No

**Claims And Evidence:**

Yes

**Requested Changes:**

As in weaknesses.

**Strengths And Weaknesses:**

Strengths:

The authors conduct extensive experiments with varying annotation amounts for each task, annotation overlaps, and dataset sizes.

The targeted problem is very interesting.

Weaknesses:

Firstly, in the paper, it seems that the authors only use FixMatch/Dense FixMatch (previous works) in semi-supervised multi-task learning, i.e. the so-called invariant/equivariant learning, rather than propose a novel method. Although the paper provides comprehensive experiments, it overall gives limited insights for future work.

Secondly, in Section 3 (Method), the usage of $\alpha$/$\mathcal{A}$ and $\alpha^{\prime}$/$\mathcal{A}^{\prime}$ is confusing. In Equation 1 and 2, $\alpha$/$\mathcal{A}$ and $\alpha^{\prime}$/$\mathcal{A}^{\prime}$ refer to invariant transformations and equivariant transformations respectively. However, in Equation 3 and 4, $\alpha$/$\mathcal{A}$ refers to both invariant transformations and equivariant transformations.

Thirdly, in Table 2 and 3, it seems that experiments (c) and (e) have different subset sizes and different overlap simultaneously. If that's the case, comparing the results with two different variables (subset sizes and overlap) does not provide useful information, so providing the trend of results with varying overlap and fixed subset sizes is a better choice. Moreover, further analysis on these results is needed, e.g. the impact of overlap or the impact of one task's subset size when another task's subset size is fixed. Merely stating that FixMatch* performs well and Dense FixMatch performs better is not sufficient.

Additionally, the writing quality of the paper has a lot of room for improvement with many formatting issues.

- Section 1: In the summary of contributions, there is a "FixMatch*", but the explanation of this term is first introduced in Section 3.
- Section 2: Redundant author name, e.g., "Wei et al. (Wei et al., 2021)"
- Equation 3: $\mathbf{x}_{i }\in \mathcal{B}_t^l \rightarrow (\mathbf{x}_{i },\mathbf{y}_{i })\in \mathcal{B}_t^l $
- Equation 4: There should be a "," at the end.
- Section 4: 800 x 800 $\rightarrow$  800 $\times$ 800
- Section 5: Fixmatch $\rightarrow$ FixMatch

---

> ### Author Response · Authors · 2024-03-17
>
> We thank the reviewer for their time and their suggestions for improvement. In the following, we attempt to comment on and give answers to the points raised as weaknesses.
>
> In this work we are not aiming at proposing any technical innovation but at framing the problem of multi-task learning on partially labeled datasets as multiple semi-supervised learning problems. To solve this, we use a well-studied and popular semi-supervised method, FixMatch, as well as an extension for equivariant tasks to still use the full spectrum of transformations, with general applicability to different tasks with different output structures and invariance or equivariance properties. Our proposed approach provides a clear benefit over the supervised baselines in this problem while using the key components of state-of-the-art semi-supervised learning methods. Given the major success in the single-task semi-supervised learning literature of the methods used and our positive results, we believe this serves as a good direction for future work in the partially-labeled multi-task learning setting.
>
> We have updated the notation of the augmentations in the Method section to provide more clarity and avoid confusion.
>
> At the end of the first paragraph of Section 4.1, we comment that experiments (c) and (e) are not directly comparable to other experiments, only to the supervised single-task and multi-task baselines. It is worth noting that each column in the experiments in this section is a different data setting, so the comparison within the column still shows that the approach is beneficial for each case. We argue that this is still interesting from a practical point of view, since a given real-world problem will have a single data setting, and there is no easy way of controlling overlap or subset size other than annotating more data.
>
> Finally, we have fixed the multiple formatting issues pointed out by the reviewer.

---

### Review · Reviewer_N1Dt · 2024-03-05

**Summary Of Contributions:**

The paper tackles the problem of multi-task learning with partial annotations, where the annotated subset of each task can have very different size and limited overlap. To this end, the paper adopts existing single-task semi-supervised learning methods, FixMatch and Dense FixMatch, and investigates their effectiveness in the partially-annotated multi-task setting. Experiments show that FixMatch and Dense FixMatch can outperform the supervised baseline in various settings regarding data amount and annotation overlap.

**Audience:**

Yes

**Claims And Evidence:**

Yes

**Requested Changes:**

- (Important) Please add some explanation why outperforming the supervised baseline is an interesting result. This may be done by showing the incompetence of existing methods, such as the ones mentioned in the last paragraph of page 3.
- (Optional) Improve the presentation of results by adding some plots.

**Strengths And Weaknesses:**

Strengths:
- The method is generic and can in principle be applied to other vision tasks.
- The investigation is extensive, covering various configurations of annotation amount and overlap that may arise in real-world applications.
- The paper provides detailed description of experiment settings and hyperparameters.

Weaknesses:
- The main takeaway of the paper is that FixMatch and Dense FixMatch can outperform the supervised baseline in multi-task learning with partial annotations. However, I am not quite sure whether this is an interesting result. The paper mentions a few other works for partially-annotated multi-task learning. Are these methods not able to outperform the supervised baseline?
- Most results are presented in large tables, making it hard to grasp the key message from the results. I think having some figures to visualize key results can be beneficial.

---

> ### Author Response · Authors · 2024-03-17
>
> We thank the reviewer for their time, for their positive comments, as well as for raising their perceived weaknesses and making suggestions for improvement. We comment on the weaknesses next.
>
> We compare to the basic approach outlined in Kokkinos (2017) of zeroing out the loss for a task a sample is not annotated for, which we consider the supervised baseline since it only uses the samples labeled for each task for learning it. We have now made this explicit in the experiments section, “Common settings” paragraph. Other prior work uses distillation techniques in alternate training steps to tackle the problem (Kim et al., 2018; Nekrasov et al., 2019), without explicitly framing it as semi-supervised learning. Seeing these methods from the semi-supervised learning perspective shows their approach is based on standard self-training with distillation, missing the key component of consistency regularization to perturbations that made FixMatch so successful. Similarly, methods tackling the problem based on adversarial learning are mentioned but this approach has fallen out of favor in the semi-supervised learning literature due to the much superior performance of consistency regularization and self-training. Again, we did not compare to these alternative approaches to keep the focus on the proposed approach and the amount of work for this paper reasonable. *Note that we do not claim this approach is better than these potential alternatives*, only that the one we propose provides a clear benefit over the supervised baselines while using the key components of state-of-the-art semi-supervised learning methods, which we believe is a good direction to build upon for future work given its major success in the single-task literature.

---

### Review · Reviewer_8Et9 · 2024-03-13

**Summary Of Contributions:**

This paper addresses the challenge of multi-task learning, aiming to handle various tasks within a unified model. Specifically, the focus is on partially labeled datasets. To efficiently leverage all available data for the tasks at hand, the paper introduces the method of invariant/equivariant semi-supervised learning which is developed on two former works FixMatch and Dense FixMatch.

**Audience:**

Yes

**Broader Impact Concerns:**

/

**Claims And Evidence:**

No

**Requested Changes:**

please see weakness

**Strengths And Weaknesses:**

While the paper explores intriguing tasks, I have identified the following concerns:

1. The writing could benefit from improvement, especially regarding the use of "invariant/equivariant" terms. It is unclear what the formal definitions for invariant and equivariant semi-supervised learning are. The distinction between these terms is not clearly outlined. While discussions are provided in the related works and method sections, the abstract and introduction lack clarity on these concepts. Additionally, the introduction of new terminology for learning paradigms, such as FixMatch and Dense FixMatch, without explicit definitions, could be confusing.

2. The technical novelty appears limited. The paper introduces FixMatch and Dense FixMatch, attributing invariance and equivariance to each, respectively. However, the contributions seem incremental, and the proposed combined training method leveraging both invariance and equivariance lacks substantial innovation. Moreover, the paper does not directly address the multi-task learning issue.

3. The experiments are not comprehensive. Specifically, there are two notable shortcomings: 1) the study focuses only on two subtasks, namely segmentation and object detection; 2) the comparison is limited to the mentioned FixMatch and Dense FixMatch methods, lacking a broader evaluation of alternative approaches.

---

> ### Author Response · Authors · 2024-03-17
>
> We thank the reviewer for their time and for raising multiple weaknesses, which we attempt to comment on next.
>
> First, we define formally what we refer to as invariant and equivariant learning in the Method section. We have updated the notation there to avoid confusion following the comments of another reviewer. We introduce the invariant and equivariant learning concepts and give a high-level explanation of them in both the Introduction and Related work. Both FixMatch and Dense FixMatch are prior art which we cite and explain in both the Introduction, Related Work, and Method sections. Moreover, we discuss how we interpret these two methods as invariant and equivariant learning. We have touched the language a bit to hopefully make that more clear in different places. We would like to ask the reviewer to point out how they perceive there is a lack of clarity around this so we can improve it further.
>
> Second, we want to clarify that in this work we are not aiming at proposing any technical novelty. Instead, the novelty is in framing the problem of multi-task learning on partially labeled datasets as multiple semi-supervised learning problems and tackling them using methods that include the key components of state-of-the-art semi-supervised learning, which we interpret as invariant/equivariant learning, namely FixMatch and its Dense FixMatch.
>
> We run multiple experiments with two different datasets, also simulating different real-world partially labeled multi-task problems in terms of the relation between the annotations for the different tasks, with different sizes and overlaps. Therefore, we argue that the experiments are indeed comprehensive from the data setting point of view, despite the shortcomings raised by the reviewer, which we will try to address below. Since we are tackling a problem that arises when simultaneously learning multiple tasks with partially labeled annotations, we argue that we do address multi-task learning. We ask the reviewer to clarify why they think the paper does not address the multi-task learning issue.
>
> Regarding using only two tasks in the multi-task learning problems considered in the experiments, we understand more tasks would offer stronger results. However, we chose to use only two tasks since it is the simplest multi-task learning setting, and we thought it would be important to have a simple setting so that the results were easier to analyse and read. We decided to focus more on choosing different tasks, finally going with two common tasks in the computer vision literature with different output structures, semantic segmentation with dense outputs and object detection with sparse outputs, to show the general applicability of the method from this perspective. Other works restrict themselves to only dense tasks (Li et al., 2022), even if considering a larger number. We comment on this in the last paragraph of the Related Work.
>
> As pointed out by the reviewer, we only compare our proposed approaches to supervised baselines which only use the samples labeled for each task for learning it, which is the basic approach outlined in Kokkinos (2017) of zeroing out the loss for a task a sample is not annotated for. We have now made this explicit in the experiments section, “Common settings” paragraph. Other prior work uses distillation techniques in alternate training steps to tackle the problem (Kim et al., 2018; Nekrasov et al., 2019), without explicitly framing it as semi-supervised learning. Seeing these methods from the semi-supervised learning perspective shows their approach is based on standard self-training with distillation, missing the key component of consistency regularization to perturbations that made FixMatch so successful. Similarly, methods tackling the problem based on adversarial learning are mentioned but this approach has fallen out of favor in the semi-supervised learning literature due to the much superior performance of consistency regularization and self-training. Again, we did not compare to these alternative approaches to keep the focus on the proposed approach and the amount of work for this paper reasonable. *Note that we do not claim this approach is better than these potential alternatives*, only that the one we propose provides a clear benefit over the supervised baselines while using the key components of state-of-the-art semi-supervised learning methods, which we believe is a good direction to build upon for future work given its major success in the single-task literature.

---

### Decision · Action_Editor_3Cpz · 2024-04-17

**Recommendation:** Reject

**Comment:**

The paper received mixed reviews, with two reviewers leaning towards rejection and one towards acceptance. It's clear that the paper did not aim to propose a new algorithm; instead, the authors focused on the partially-annotated multi-task setting and conducted an extensive empirical study of existing single-task semi-supervised learning methods. This approach is indeed valid. However, the reviewers criticized the need for further development in the experimental design to uncover deeper insights. For example, it would be beneficial if the authors could explain how and why the chosen single-task semi-supervised learning methods perform better. Even if the explanation is empirical, it could provide more insightful contributions and valuable lessons to the field. Also, a comparison with a few other works for partially-annotated multi-task learning will make the experimental part more complete.

**Audience:**

The main takeaway from the paper is that FixMatch and Dense FixMatch can surpass the supervised baseline in multi-task learning with partial annotations. This finding could be valuable for practitioners; however, the paper's broader insights might be limited. It remains unclear how and why the chosen methods, FixMatch and Dense FixMatch, perform better, as the paper does not fully elucidate the underlying mechanisms. Additionally, the paper references several other approaches to partially-annotated multi-task learning but does not clarify whether these methods also surpass supervised baselines. This omission could leave the audience curious about the comparative effectiveness of these methods, overshadowing the concrete contributions of the study. Such unanswered questions may detract from the meaningful conclusions that could otherwise be drawn from the research.

**Claims And Evidence:**

The paper evaluates the effectiveness of established single-task semi-supervised learning methods—FixMatch and Dense FixMatch—in a partially-annotated multi-task setting. Comprehensive experiments are conducted on the Cityscapes and BDD100K datasets for semantic segmentation and object detection tasks. The findings reveal that both invariant and equivariant learning surpass supervised baselines, especially in scenarios with limited annotations. However, reviewers have criticized the presentation of results. They suggest it would be more insightful to demonstrate trends with varying levels of overlap and fixed subset sizes. Additionally, there is a need for deeper analysis of these results, such as exploring the impact of overlap or the effects of one task's subset size when another's remains constant. The paper's current content, asserting merely that FixMatch* performs well and Dense FixMatch performs better, is deemed insufficient.

**Resubmission Of Major Revision:**

The authors may consider submitting a major revision at a later time.